# Prognostic Implications of Portal Venous Circulating Tumor Cells in Resectable Pancreatic Cancer

**DOI:** 10.3390/biomedicines10061289

**Published:** 2022-05-31

**Authors:** Young Hoon Choi, Tae Ho Hong, Seung Bae Yoon, In Seok Lee, Myung Ah Lee, Ho Joong Choi, Moon Hyung Choi, Eun Sun Jung

**Affiliations:** 1Department of Internal Medicine, College of Medicine, The Catholic University of Korea, Seoul 06591, Korea; crzyzs@naver.com (Y.H.C.); isle@catholic.ac.kr (I.S.L.); angelamd@catholic.ac.kr (M.A.L.); 2Department of Surgery, College of Medicine, The Catholic University of Korea, Seoul 06591, Korea; gshth@catholic.ac.kr (T.H.H.); hopej0126@catholic.ac.kr (H.J.C.); 3Department of Radiology, College of Medicine, The Catholic University of Korea, Seoul 06591, Korea; cmh@catholic.ac.kr; 4Department of Hospital Pathology, College of Medicine, The Catholic University of Korea, Seoul 06591, Korea; esjung@catholic.ac.kr

**Keywords:** pancreatic cancer, circulating tumor cells, biomarkers, survival

## Abstract

Circulating tumor cells (CTCs) are a promising prognostic biomarker for cancers. However, the paucity of CTCs in peripheral blood in early-stage cancer is a major challenge. Our study aimed to investigate whether portal venous CTCs can be a biomarker for early recurrence and poor prognosis in pancreatic cancer. Patients who underwent upfront curative surgery for resectable pancreatic cancer were consecutively enrolled in this prospective study. Intraoperatively, 7.5 mL of portal and peripheral blood was collected, and CTC detection and identification were performed using immunofluorescence staining. Peripheral blood CTC sampling was performed in 33 patients, of which portal vein CTC sampling was performed in 28. The median portal venous CTCs (2.5, interquartile ranges (IQR) 1–7.75) were significantly higher than the median peripheral venous CTCs (1, IQR 0–2, *p* < 0.001). Higher stage and regional lymph node metastasis were related with a larger number of CTCs (≥3) in portal venous blood. Patients with low portal venous CTCs (≤2) showed better overall (*p* = 0.002) and recurrence-free (*p* = 0.007) survival than those with high portal venous CTCs (≥3). If validated, portal CTCs can be used as a prognostic biomarker in patients with resectable pancreatic cancer.

## 1. Introduction

Pancreatic ductal adenocarcinoma is one of the leading causes of cancer-related deaths in both sexes worldwide, and its prevalence and incidence rates are increasing [1]. Despite advances in multidisciplinary approaches, pancreatic cancer remains a devastating malignancy with a five-year survival rate less than 5% [2]. Even in resectable pancreatic cancer patients undergoing curative resection, more than 40% of patients experienced recurrence or metastasis within one year after surgery [3]. For curative treatment of pancreatic cancer, personalized treatments using precise prognostic or predictive biomarkers are essential.

Circulating tumor cells (CTCs) are among the cancer-derived materials shed from primary tumors into systemic circulation. CTCs are considered as minimally invasive biomarkers for tumor burden and precursors for metastasis [4]. Previous studies have demonstrated that CTCs in the peripheral circulation can serve as prognostic biomarkers for advanced breast, prostate, and colon cancers [5,6,7]. In pancreatic cancer, efforts have been made to apply CTC quantification to prognosis, mainly for metastatic or advanced diseases. However, the rarity of CTCs in the peripheral venous blood of patients with non-metastatic cancer limits its clinical use as a predictive or prognostic biomarker. Furthermore, the detection of rare CTCs in peripheral blood of early-stage cancer is challenging with the currently available techniques. The paucity of CTCs in the peripheral blood of pancreatic cancer patients is attributed to hepatic sequestration [8]. The bloodstream of the pancreas flows through the portal vein into the liver, and the hepatic filtration effect can reduce CTCs in the peripheral circulation. For this reason, liver, the first-pass organ for pancreas venous blood drainage via the portal vein, is the most frequent site of distant metastasis of pancreas cancer. Theoretically, CTCs will be more commonly detected in portal than in peripheral blood of patients with pancreatic cancer.

The main aims of this study were to evaluate the correlation between the number of portal venous CTCs and long-term clinical outcome and to verify the clinical application of portal liquid biopsy in resectable pancreatic cancer patients.

## 2. Materials and Methods

### 2.1. Study Participants

Patients who were scheduled to undergo curative resection for pancreatic cancer from September 2017 to June 2019 at a single tertiary hospital were consecutively enrolled in this prospective study. Only patients pathologically confirmed with pancreatic ductal adenocarcinoma using endoscopic ultrasound (EUS)-guided fine needle biopsy before surgery were enrolled. Exclusion criteria for the study were (1) pancreatic cancer with major vessel involvement or prominent metastasis at the time of diagnosis and (2) pancreatic cancer first treated with neoadjuvant chemotherapy.

Treatment decisions were made in multidisciplinary meetings of expert gastroenterologists, oncologists, surgeons, radiologists, and pathologists. If a case was determined as resectable at the meeting, upfront surgery combined with active lymphadenectomy was performed. Neoadjuvant chemotherapy or chemoradiotherapy was not considered in resectable pancreatic cancer during the study period. To reduce local recurrence and to analyze the appropriate nodal status, surgeons performed aggressive lymphadenectomy and tried to retrieve at least 12 regional lymph nodes during surgery. Informed consent was obtained from all patients, and the institutional review board approved this study (KC17TESI0448).

### 2.2. Clinical Data and Sample Collection

Demographic, radiologic, and laboratory data including serum carbohydrate 19-9 (CA19-9) level were collected. The 8th edition of the American Joint Committee of Cancer/Union for International Cancer Control (AJCC/UICC) TNM classification was used for clinical tumor staging. Two expert surgeons (THH and HJC) with more than 10 years of experience inspected and palpated the liver and peritoneal cavity to identify any possible metastasis. The surgeons also assessed whether the tumor invaded the major vessels. 

Intraoperatively, 7.5 mL of blood was collected from the portal vein by direct puncture with a syringe with a 21-gauge needle before manipulation of the tumor. At the time of portal venous blood collection, 7.5 mL of peripheral venous blood was also collected. All blood samples were stored in BD Vacutainer tubes containing anti-coagulant citrate dextrose solution (BD Biosciences, Franklin Lakes, NJ, USA).

### 2.3. CTC Enrichment

CTC enrichment was processed within 4 h after sampling to minimize cell loss and processing failure. Blood samples were incubated for 20 min with 20 µg/µL of antibody cocktail against red blood cells and white blood cells (WBCs) from the specialized CTC isolation kit (Cat no. CIKW10; Cytogen, Seoul, Korea). The samples were mixed with preactivation buffer and underwent density gradient centrifugation at 400× *g* for 30 min at room temperature. The cell suspension containing CTCs was collected and gradually diluted with phosphate buffered saline. The diluted cell suspension was passed through a high-density microporous chip (SMART BIOPSY^TM^ Cell Isolator; Cytogen Inc., Seoul, Korea) to obtain non-leukocyte, nucleated cells [9]. For immunofluorescent staining, the isolated cells were retrieved and fixed in 4% paraformaldehyde for 5 min at room temperature.

### 2.4. CTC Identification Using Immunofluoresecence Staining

Enriched cells were harvested and fixed on a microscope slide using Shandon Cytospin™ 4 (Thermo Fisher Scientific, Waltham, MA, USA). After blocking with 1% bovine serum albumin in phosphate buffered saline for 30 min, cells were incubated with monoclonal antibodies (mAbs) against EpCAM (dilution 1:50; Cell Signaling Technology, Danvers, MA, USA), cytokeratin (CK, dilution 1:500, BD Biosciences, San Diego, CA, USA), vimentin (dilution 1:125, Cell Signaling, MA, USA), and CD45 (dilution 1:100; Agilent, Santa Clara, CA, USA). Nuclei were stained with 4′,6-diamidino-2-phenylindole (DAPI; Abcam, Cambridge, CB2, UK), and cells were examined under a fluorescence microscope (SMART BIOPSY^TM^ Cell Image Analyzer; Cytogen Inc., Seoul, Korea) with a 400× objective.

Quantification was performed by a single human observer. Total cells were counted by DAPI staining, and WBCs were identified by CD45 staining. Captured cells were determined to be positive if the cells had intact morphology and were greater than 15 μm in size, DAPI positive, CD45 negative, and EpCAM or CK positive (DAPI^+^, CD45^−^, EpCAM/CK^+^). Epithelial-type CTCs (E-CTCs) were defined as those without expression of vimentin (DAPI^+^, CD45^−^, EpCAM/CK^+^, vimentin^−^), while mesenchymal-type CTCs (M-CTCs) were defined as those expressing vimentin (DAPI^+^, CD45^−^, EpCAM/CK^+^, vimentin^+^).

### 2.5. Follow-Up Strategy and Survival Analysis

If patients were eligible for chemotherapy, gemcitabine- or 5-FU-based adjuvant chemotherapy was administered to resected patients for 3 to 6 months, regardless of pathologic stage. All patients were planned to be followed-up at 3-month intervals for 1 year after surgery and at 6-month intervals thereafter. Palliative chemotherapy with FOLFIRINOX or gemcitabine plus albumin-bound paclitaxel was administered to patients experiencing recurrence during follow-up. Overall survival (OS) was calculated from the date of surgery to the last date of follow-up or the date of death. Recurrence-free survival (RFS) was calculated from the date of operation to the last date of follow-up or the date of first documented recurrence on imaging studies. The primary study endpoint was the relation between portal CTC count and survival. The secondary endpoints were (i) differences between portal and peripheral CTC counts, (ii) correlations between CTC counts and patient or tumor characteristics, and (iii) differences of clinical outcomes between patients with E-CTCs and M-CTCs.

## 3. Results

### 3.1. Study Population

Between September 2017 and June 2019, 33 patients were enrolled prospectively into the study. Intraoperatively, liver metastases were found in three patients, and peritoneal seeding was found in one. One patient did not undergo curative resection because superior mesentery artery invasion was found during surgery. Peripheral blood CTC sampling was performed in 33 patients, and portal vein CTC sampling was performed in 28 (excluding 5 patients) (Figure 1).

The baseline characteristics of the study patients (N = 33) are shown in Table 1. The mean age was 63.1 ± 11.6 years, and the study group included 14 (42.4%) males and 19 (57.6%) females. Twenty-eight patients (84.8%) underwent curative surgery and five patients received palliative chemotherapy for the first treatment. Among the 28 surgical cases, pancreaticoduodenectomy and distal pancreatectomy were performed in 24 (85.7%) and 4 (14.3%) cases, respectively. Surgical margins were negative (R0) in 24 (82.1%) patients, and adjuvant or palliative chemotherapy was administered after surgery to 27 (96.4%) patients. According to the AJCC/UICC 8th edition, 9 (27.3%), 14 (42.4%), 6 (18.2%), and 4 (12.1%) patients were classified into stage I, II, III, and IV, respectively.

### 3.2. CTC Identification

We counted CTCs using a four-color staining protocol of DAPI (blue) for nucleated cells, CD45 (red) as a leukocyte marker, EpCAM or CK (yellow) as an epithelial marker, and vimentin (green) as a mesenchymal marker. Representative images of E-CTCs, M-CTCs, and leukocytes are shown in Figure 2. If two-thirds or more of the number of CTCs were E-CTCs or M-CTCs, the patient was classified as E-CTC dominant type (E-type) or M-CTC dominant type (M-type), respectively.

### 3.3. Comparison of CTC Number between Peripheral and Portal Venous Blood

CTCs were detected in peripheral and portal venous blood in 75.8% (25/33) and 82.1% (23/28) of cases, respectively (*p* = 0.757, Figure 3A). In the 28 patients undergoing sampling of both peripheral and portal blood, the median portal venous CTCs (2.5, interquartile range (IQR) 1–7.75) were significantly higher than the median peripheral venous CTCs (1, IQR 0–2, *p* < 0.001, Figure 3B). The number and subtypes of CTCs in peripheral or portal venous blood of individual patients are described in Appendix A.

### 3.4. Factors Associated with a Higher Number of CTC

The number of CTCs in peripheral blood was not associated with any patient- or tumor-related factor (Table 2). Higher stage and regional lymph node metastasis were related to a higher number of CTCs in portal venous blood. The majority of patients (57.1%, 8/14) with low portal venous CTCs (≤2) were stage I, while all except one patient (92.9%, 13/14) with high portal venous CTCs (≥3) were stage II or III. Regional lymph node metastasis was significantly higher in high portal venous CTC (≥3) patients than in low portal venous CTC (≤2) patients (85.7 vs. 38.5%, *p* = 0.018).

### 3.5. Survival Analysis

The median duration of follow-up in the operated patients was 23.2 months. The Kaplan–Meier survival analysis showed that patients with low portal venous CTCs (≤2) had better.

OS than those with high portal venous CTCs (≥3) (Figure 4A, *p* = 0.002 by the log-rank test). The median OS was not achieved in low portal venous CTC (≤2) patients and was 16.5 months in high portal venous CTC (≥3) patients. Patients with low portal venous CTCs (≤2) also showed better RFS than those with high portal venous CTCs (≥3) (Figure 4B, *p* = 0.007, median RFS 13.4 vs. 7.0 months). The number of peripheral venous CTCs was not related to OS and RFS after resection of pancreatic cancer (Figure 4C,D).

### 3.6. Comparsion according to Phenotype of CTC

In the analysis of the twenty-three patients with more than one portal venous CTC detected, most (95.7%, 22/23) showed a dominant type. The numbers of E-type and M-type patients were 9 (39.1%) and 13 (56.5%), respectively, and only one patient was unclassified. There were no significant differences in patient and tumor characteristics between E-type and M-type patients (Appendix A). No differences in OS and RFS were found between E-type and M-type patients (Appendix A).

## 4. Discussion

In this study, we evaluated the prognostic value of peripheral and portal venous CTCs in patients with resectable pancreatic cancer. The median number of CTCs in portal vein was significantly higher than that in peripheral blood. High numbers of portal CTCs were associated with regional lymph node metastasis and high TNM stages. In survival analysis, patients with a high number of portal CTCs had shorter RFS and OS than those with a low number of portal CTCs. The number of peripheral CTCs did not correlate with tumor stage or survival. 

CTCs are cells that have been shed into the bloodstream from tumors and are thought to be the seed of tumor metastasis [4]. Allard et al. found that pancreatic cancer, along with other tumors of the gastrointestinal tract, was associated with lower numbers of CTCs in the peripheral blood compared with other carcinomas, possibly due to hepatic filtration through the portal vein [10]. Based on this finding, some studies compared portal and peripheral CTCs in patients with pancreatic cancer. Two recent studies of patients with advanced or metastatic pancreatic cancer showed a higher CTC detection rate in the portal vein than in the peripheral vein, and the number of CTCs derived from the portal vein, rather than CTCs from the peripheral vein, was associated with OS [11,12]. However, studies that included only patients with resectable pancreatic cancer showed inconsistent results. Pan et al. analyzed 60 patients with resectable pancreatic cancer and showed similar results to our study in terms of differences between portal and peripheral CTC number and their association with survival [13]. In contrast, another study by Song et al. that included 32 resectable pancreatic cancer patients found no differences in portal and peripheral CTC detection rates and no association between number of portal CTCs and survival [14]. It is likely that there are several factors influencing the difference in these study results, with CTC detection rate estimated to be one of the major factors. The portal CTC detection rate was greater than 80% in some studies, including our study, in which portal CTC was significantly correlated with survival. However, the portal CTC detection rate remained in the 60% range in studies that did not show such correlation [11,12,13,14].

Factors affecting CTC detection rates, excluding tumor burden, include CTC enrichment methods and immunofluorescence staining markers used to identify CTCs [15,16]. For CTC enrichment, size-based method with a high-density microporous chip named SMART BIOPSY^TM^ was used in our study. This method has been widely used in clinical researches in various cancers these days [14,17,18,19]. In addition, using the existing data, we confirmed that the CTC capture efficiency of this method was about 85%, which was comparable with other methods, such as ISET^®^ or Cellsearch^®^ [9,20]. In three recent studies on patients with resectable pancreatic cancer, including our study, all the CTC enrichment methods used size-based filtration [13,14]. There might be differences in CTC detection performance because of the different size-based filtration methods used among studies, but the basic concepts and principles of CTC enrichment and identification process are similar among the studies. However, immunofluorescence staining markers used for CTC detection were different for each study. In our study and that of Pan et al., EpCAM, CK, and vimentin were used, whereas Song et al. only used EpCAM and Plectin-1. EpCAM, CK, and vimentin are the most commonly used molecular markers to identify CTCs and have been verified in several cancers [4]. Therefore, the absence of CK or vimentin markers in Song et al. might have contributed to the relatively low CTC detection rate around 60% [14]. 

The present study showed that the number of portal CTCs correlated with regional lymph node metastases. To our knowledge, this is the first study to demonstrate the association between the number of portal CTCs and regional lymph node metastasis in patients with resectable pancreatic cancer. Zhang et al. also reported that the number of portal CTCs was related to lymph node metastasis, but the study included patients with metastatic or locally advanced diseases, in addition to resectable pancreatic cancer [12]. Lymph node metastasis is particularly important in resectable pancreatic cancer because it is associated with early recurrence and poor clinical outcome [21]. Our study showed that a high number of portal CTCs was associated with not only regional lymph node metastasis, but also a shorter RFS, indicating early recurrence. This is important as portal CTC can be considered as a biomarker candidate that distinguishes a group of patients with resectable pancreatic cancer with predicted early recurrence, in whom neoadjuvant chemotherapy might be beneficial. Even after curative resection, the recurrence rate of pancreatic cancer reaches 85%; therefore, recently, resectable pancreatic cancer is being considered a systemic disease, and studies on neoadjuvant chemotherapy are being actively conducted [22,23]. However, most trials of neoadjuvant chemotherapy for resectable pancreatic cancer have not demonstrated improved survival [24,25,26,27]. Therefore, future studies are needed to confirm the efficacy of neoadjuvant chemotherapy in a subgroup of patients with resectable pancreatic cancer identified with high risk of early recurrence using biomarkers, such as portal CTCs. With the application of a method to safely collect portal venous CTCs through EUS-guided fine needle aspiration, it is possible to identify suitable patients for neoadjuvant chemotherapy before surgery [28,29].

Our study did not show the clinical significance of CTC phenotypes such as epithelial and mesenchymal types. Epithelial-mesenchymal transition plays an important role in metastasis, and several reports showed that mesenchymal-type CTC is associated with worse prognosis in several carcinomas [30]. The small number of subjects in this study might have obscured the differences in prognosis according to CTC phenotype. Zhang et al.’s study using Twist as a mesenchymal CTC marker reported a greater number of Twist^+^ CTCs compared with EpCAM^+^ CTCs in the portal vein [12]. Therefore, the absence of additional mesenchymal markers, such as Twist, in addition to vimentin in this study might explain the absence of a difference in prognosis according to CTC phenotype.

This study has several limitations. First, we did not perform further molecular tests for CTC identification, such as KRAS mutation, which is the most common mutation in pancreatic cancer. However, the association of cancer prognosis with CTCs identified through immunofluorescence staining based on EpCAM and CK has been proven in previous studies on pancreatic cancer [31,32]. Therefore, the results on the prognostic value of portal CTC based on immunofluorescence staining in this study are also thought to be meaningful. Further studies, including molecular tests for CTC, are needed in the future. Second, this study was conducted with a small number of patients in one tertiary medical institution. Third, since patients who received neoadjuvant chemotherapy were excluded, the clinical significance of portal CTCs cannot be directly applied to this patient group. Further studies are needed to evaluate the clinical usefulness of portal CTC in resectable pancreatic cancer patients who receive neoadjuvant chemotherapy.

## 5. Conclusions

Our study found that a high number of portal CTCs was associated with regional lymph node metastasis, shorter RFS and shorter OS in patients with resectable pancreatic cancer. If validated, portal CTC can be used as a prognostic biomarker in patients with resectable pancreatic cancer.

## Figures and Tables

**Figure 1 biomedicines-10-01289-f001:**
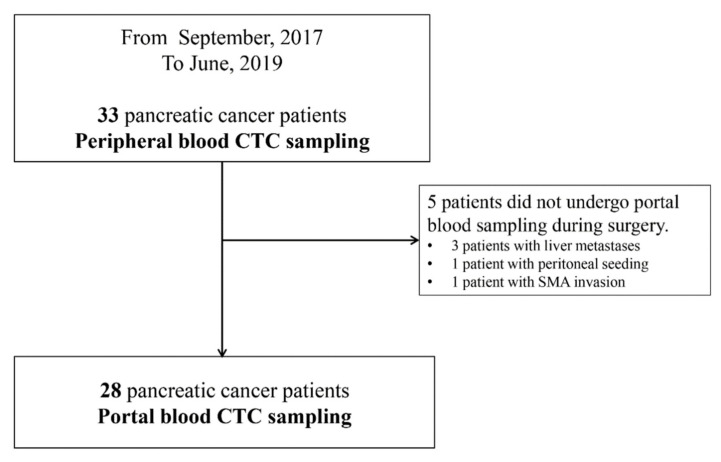
Flowchart of the enrolled patients.

**Figure 2 biomedicines-10-01289-f002:**
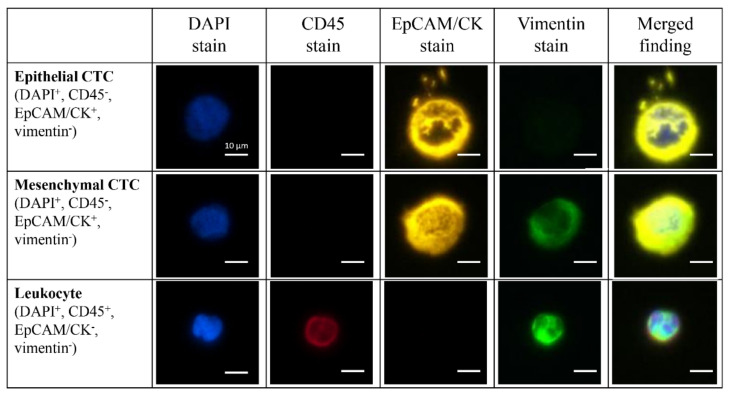
Representative images of epithelial-type CTCs (E-CTC), mesenchymal-type CTCs (M-CTC), and leukocytes (scale bar: 10 μm). CTCs are greater than 15 μm, DAPI-positive, CD45-negative, and EpCAM- or CK-positive (DAPI^+^, CD45^−^, EpCAM/CK^+^). E-CTCs were defined as CTCs without expression of vimentin (DAPI^+^, CD45^−^, EpCAM/CK^+^, vimentin^−^), and M-CTCs were defined as CTCs that express vimentin (DAPI^+^, CD45^−^, EpCAM/CK^+^, vimentin^+^). Leukocytes are stained for CD45.

**Figure 3 biomedicines-10-01289-f003:**
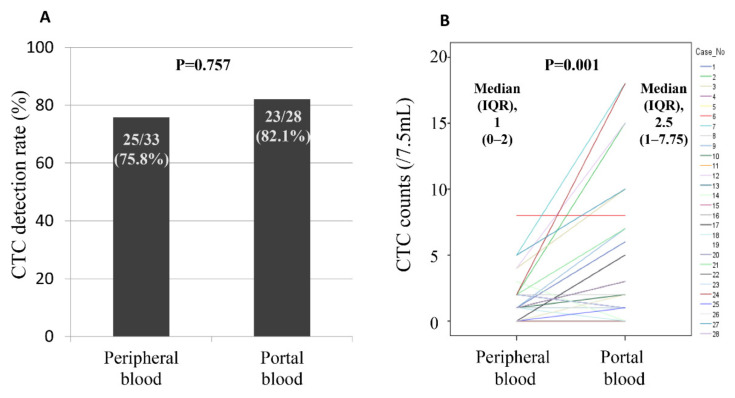
Comparison of CTC detection rate and number between peripheral and portal venous blood. (**A**) CTCs were detected in 75.8% (25/33) and 82.1% (23/28) of samples of peripheral and portal venous blood (*p* = 0.757), respectively. (**B**) Median CTC number in the portal vein was significantly higher than that in the peripheral blood (median, 2.5 vs. 1 cells/7.5 mL of blood, *p* < 0.001).

**Figure 4 biomedicines-10-01289-f004:**
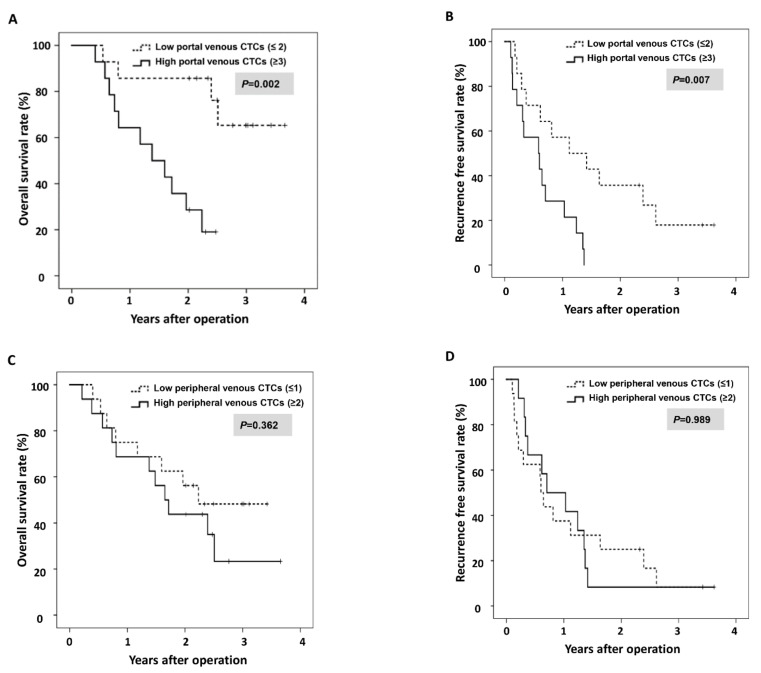
Survival analyses in patients by level of CTCs. (**A**,**B**) Kaplan–Meier survival analysis showed that patients with low portal venous CTC (≤2) had better overall survival (OS) and progression-free survival (PFS) compared with those with high portal venous CTCs (≥3) (*p* = 0.002 and *p* = 0.007, respectively). (**C**,**D**) The number of peripheral venous CTCs was not associated with OS and RFS after resection of pancreatic cancer.

**Table 1 biomedicines-10-01289-t001:** Patient and tumor characteristics (N = 33).

Parameters	Data
**Characteristics**	
Age, years	63.1 ± 11.6
Sex, male (%)	14 (42.4%)
CA 19-9, median (IQR), U/mL	141 (28–837)
Undergoing curative surgery (%)	28 (84.8%)
**Among operated cases (n = 28)**	
Pancreaticoduodenectomy (%)	24 (85.7%)
Distal pancreatectomy (%)	4 (14.3%)
R0 resection	23 (82.1%)
Adjuvant or palliative chemotherapy	27 (96.4%)
**Among non-operated cases (n = 5)**	
Palliative chemotherapy	5 (100%)
**Stage by AJCC 8th edition**	
Stage I	9 (27.3%)
Stage II	14 (42.4%)
Stage III	6 (18.2%)
Stage IV	4 (12.1%)

**Table 2 biomedicines-10-01289-t002:** Factors associated with low vs. high number of CTCs from peripheral and portal venous blood.

Parameters	Peripheral Venous CTCs (N = 33)			Portal Venous CTCs (N = 28)		
	Low (≤1) (n = 16)	High (≥2) (n = 17)	*p* Value	Low (≤2) (n = 14)	High (≥3) (n = 14)	*p* Value
**Patient Characteristics**						
Age, mean ± SD, years	63.6 ± 13.3	62.6 ± 10.2	0.802	62.9 ± 11.6	61.5 ± 12.2	0.753
Sex, male (%)	7 (43.8%)	7 (41.2%)	0.881	7 (50.0%)	6 (42.9%)	0.705
CA 19-9, median (IQR), (U/mL)	109 (22–649)	326 (41–978)	0.494	73 (19–485)	354 (61–866)	0.141
**Tumor Characteristics**						
Stage			0.124			0.016
Stage I	6 (37.5%)	3 (17.6%)		8 (57.1%)	1 (7.1%)	
Stage II	8 (50.0%)	6 (35.5%)		4 (28.6%)	10 (71.4%)	
Stage III	2 (12.5%)	4 (23.5%)		2 (14.3%)	3 (21.4%)	
Stage IV	0 (0%)	4 (23.5%)		0 (0%)	0 (0%)	
Primary tumor size, mean ± SD, cm	3.6 ± 1.4	3.7 ± 1.2	0.879	3.7 ± 1.6	3.7 ± 1.1	0.978
Regional LN involvement (%)	NA	NA	NA	5 (38.5%)	12 (85.7%)	0.018
Tumor differentiation			0.527			0.328
Well-differentiated (%)	5 (31.3%)	5 (29.4%)		6 (42.9%)	2 (14.3%)	
Moderately differentiated (%)	7 (43.8%)	10 (58.8%)		6 (42.9%)	8 (57.1%)	
Poorly differentiated (%)	4 (25.0%)	2 (11.8%)		2 (14.3%)	4 (28.6%)	

NA, not available.

## Data Availability

The data are not publicly available due to private information of patients.

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
