# Peer review of "Prognostic Implications of Portal Venous Circulating Tumor Cells in Resectable Pancreatic Cancer"

_biomedicines, 2022, doi:10.3390/biomedicines10061289_

Round 1

Reviewer 1 Report

This is an interesting article in which authors compared portal and peripheral blood to identify CTCs. While I found the study design and use of clinical samples interesting, there are some major issues which must be addressed

1- Why is there no molecular test for CTC identification? It is well known that IHC is not enough to score a cell as CTC as it can be another non hematologic cell in circuriton! The authors need to address this issue!

2- Did authors check the waste channels to see how many putative CTCs they lose with their system?

3- â€‹any samples from lymph nodes for CTCs?​

Reviewer 2 Report

The authors report original data on the prognostic value of circulating tumor cells detection in the portal vein of patients who had surgery for pancreatic cancer. The technique used for CTC detection is based on filtration of blood cells followed by immunofluorescence staining. The principles of this approach for CTC detection are well known and largely published. The results provided by the present study have been previously reported in the literature but not with this technique.

The paper is well written and the discussion well balanced, including the small number of patients.

Have the authors compared their technique of detection with others techniques based on filtration as ISET or technique based on immunoselection as CellSearch?

Round 2

Reviewer 1 Report

the revised version looks good and it can be accepted